# Polyethylene Glycol Functionalized Silicon Nanowire Field-Effect Transistor Biosensor for Glucose Detection

**DOI:** 10.3390/nano13030604

**Published:** 2023-02-02

**Authors:** Yan Zhu, Qianhui Wei, Qingxi Jin, Gangrong Li, Qingzhu Zhang, Han Xiao, Tengfei Li, Feng Wei, Yingchun Luo

**Affiliations:** 1School of Chemical Engineering, Guizhou Minzu University, Guiyang 550025, China; 2State Key Laboratory of Advanced Materials for Smart Sensing, GRINM Group Co., Ltd., Beijing 100088, China; 3GRIMAT Engineering Institute Co., Ltd., Beijing 101407, China; 4GRINM (Guangdong) Institute for Advanced Materials and Technology, Foshan 528051, China; 5Advanced Integrated Circuits R&D Center, Institute of Microelectronic of the Chinese Academy of Sciences, Beijing 100029, China

**Keywords:** silicon nanowire, field-effect transistor biosensor, glucose detection, polyethylene glycol

## Abstract

Accurate monitoring of blood glucose levels is crucial for the diagnosis of diabetes patients. In this paper, we proposed a simple “mixed-catalyzer layer” modified silicon nanowire field-effect transistor biosensor that enabled direct detection of glucose with low-charge in high ionic strength solutions. A stable screening system was established to overcome Debye screening effect by forming a porous biopolymer layer with polyethylene glycol (PEG) modified on the surface of SiNW. The experimental results show that when the optimal ratio (APTMS:silane-PEG = 2:1) modified the surface of silicon nanowires, glucose oxidase can detect glucose in the concentration range of 10 nM to 10 mM. The sensitivity of the biosensor is calculated to be 0.47 μAcm^−2^mM^−1^, its fast response time not exceeding 8 s, and the detection limit is up to 10 nM. This glucose sensor has the advantages of high sensitivity, strong specificity and fast real-time response. Therefore, it has a potential clinical application prospect in disease diagnosis.

## 1. Introduction

Diabetes mellitus is a common, chronic, noninfectious disease caused by abnormal insulin secretion [1]. Diabetes can cause a series of diseases, such as cardiovascular disease, cerebrovascular disease, increasing the risk of cancer, etc., thus, harm human health. [2]. Blood glucose level is an index for clinical diagnosis and monitoring of diabetes. The accurate detection of blood glucose has clinical significance for the diagnosis and control of blood glucose. In recent decades, with the development of various sensing technologies, many methods for glucose detection have been reported, such as electrochemistry [3], spectrometric methods [4], colorimetric methods [5], etc. There are a lot of problems in these methods, such as a series of sample preparation, difficulty in miniaturization and labeling. At present, the detection of blood glucose is focussed to continuous monitoring of blood glucose levels through in vivo detection. Therefore, implantable nanoscale sensing devices are more meaningful to glucose home test.

Semiconductor nanowire field-effect transistor (FET) biosensors have the advantages of easy miniaturization, high sensitivity, fast response, label-free detection, commercial and large-scale application, and mass production [6,7]. They are already widely used to detect viruses [8], proteins [9], DNA [10], antigens [11], gas [12], etc., which provides the possibility to develop high performance and low cost implantable glucose biosensor. Due to the remarkable sensitivity, selectivity, real-time and label-free detection capabilities of SiNW-FET, it has attracted much attention in ultrasensitive biomarker detection [13]. In SiNW-FET-based biosensors, targeted receptors can be immobilized on the surface of silicon nanowires to functionalize the surface and selectively bind to targeted analytes. In order to achieve specificity and versatility, surface of biosensor must be functionalized by receptors and other nanostructures [14,15]. It leads to the increase (or decrease) of the charge, which in turn leads to the increase (or decrease) of the conductance. Unfortunately, SiNW-FET is limited to the detection of analytes with a certain charge, and the specific detection of weakly charged or uncharged analytes remains a challenge [16]. It has been reported that the method of modifying glucose oxidase on the surface of SiNW-FET biosensors has been successfully used to measure uncharged low-molecular-weight molecular glucose with detection limits down to μM levels [17,18]. In addition, the limitation of low charge analytes is difficult to detect. Electrical signals detected by FET-based biosensors in high ionic strength solution (serum) which are interfered by the Debye screening effect. Meanwhile, the Debye screening effect can greatly affect the sensitivity including other properties of biosensors. Simply put, there is significant untapped opportunity in the FET biosensor for the clinical detection of glucose.

In this paper, we proposed a simple 3-(aminopropyl) trimethoxysilane/polyethylene glycol/silicon nanowire field-effect transistor (APTMS/PEG/SiNW-FET) biosensor that enabled direct detection of low charge glucose molecules in high ionic strength solutions. We demonstrate a “mixed-catalyzer layer”, co-modifying SiNW-FETs with PEG and Glucose oxidase (GOD). The function of the “mixed-catalyzer layer” is to adjust the intermolecular distance of the GOD probes to achieve the optimal combination of GOD and glucose. On the other hand, signal change of GOD-glucose complex is improved by the increase of Debye length under the condition of high ionic strength. The biosensors have a higher recognition ratio toward glucose. In particular, the APTMS/PEG/SiNW-FET biosensor modified with APTMS:silane-PEG = 2:1 showed excellent performance in the detection of glucose content at low levels down to 10 nM and a wide dynamic range (10 nM ~ 10 mM). This indicates that the sensor has high practical and scientific value in future in vitro diagnostic applications.

## 2. Materials and Methods

Reagents and Chemicals

Glucose oxidase (GOD), Bovine serum albumin (BSA), 3-(aminopropyl) trimethoxysilane (APTMS), silane-PEG (10 kDa), glucose, uric acid (UA), ascorbic acid (AA) were purchased from Aladdin (Shanghai, China). Sodium hydrogen phosphate (Na_2_HPO_4_, 99%), sodium dihydrogen phosphate anhydrous (NaH_2_PO_4_, 99%) and analytical grade ethanol (C_2_H_5_OH, 99.5%) were purchased from Alfa Aesar (Shanghai, China). Glutaraldehyde (GA) was purchased from Innochem. All chemicals were used without further purification. Deionized water was used in all this experiment.

Fabrication of SiNW-FET sensor. The SiNW-FET chips used in this study are similar to those used in recent publications by our group [19]. They were fabricated based on the advanced 200 mm CMOS platform.

Surface modification of SiNW-FET biosensor. After fabricating the sensor, bioreceptors were modified on its surface for detecting target analytes. The preparation of APTMS/PEG/SiNW-FET biosensors was as follows. Firstly, the SiNW-FET device was cleaned with acetone, ethanol and deionized water to remove impurities, and the oxygen plasma was pretreated for 2 min in order to prepare the surface modification. The biosensors were then immersed in ethanol solutions containing different ratios of APTMS and silane-PEG for 30 min. The concentration ratio was 1:1, 1:2, 2:1, in which the concentration of silane-PEG was 0.2 mM. The APTMS and silane-PEG residues were then washed with ethanol. The device was treated at 120 °C for 20 min to ensure the formation of amine-terminated SiNW-FET surfaces. It was then incubated with 2.5% glutaraldehyde (GA) solution for 1 h at room temperature and rinsed several times with PBS solution to remove residual GA. Next, the FET device was immersed in 1.0 mg/mL GOD solution, placed in a 4 °C refrigerator and incubated overnight. After incubation, the FET sensor was rinsed several times with a small amount of PBS solution to remove residual GOD and dry at room temperature for 5–10 min. In order to prevent non-specific adsorption, the above-mentioned devices were incubated with 1.0 mg/mL BSA solution prepared with PBS solution at 4 °C for 12 h, and then the FET was washed several times with a small amount of PBS solution to obtain the sensor for the glucose test.

Characterization of morphology and structure. The top view of the device structure was observed using an S-5500 scanning electron microscopes (SEM, Hitachi, Tokyo, Japan). The cross section of the final device was analyzed using transmission electron microscopy (TEM, FEI Talos, Brno, Czech Republic) and energy-dispersive X-ray spectroscopy (EDX, FEI Talos, Brno, Czech Republic). Atomic force microscope (AFM, Park System NX- HDM, Seoul, Republic of Korea) was used to demonstrate the successful surface modification of the SiNW-FET device.

Measurements of detection performance of SiNW-FET biosensor. The electrical performance of liquid-gated APTMS/PEG/SiNW-FET was measured by a 4200 semiconductor parameter analyzer and four-point probe station (Keithley 4200-SCS, Santa Rosa, CA, USA). To evaluate the performance of the device, PBS solution was first dropped on the SiNW surface as a blank sample for testing. Then, PBS solution containing different concentrations of glucose was dropped on the surface of SiNW from low concentration to high concentration, and the silver probe was tested in PBS solution as a gate electrode. In order to eliminate the interference, the silver probe was rinsed twice with PBS solution before the next experiment. Transfer characteristic curves (I_DS_−V_G_) were obtained when V_DS_ was fixed at 2 V and V_G_ was set to scan in the range of −5 V to 1 V. Real-time response curves (I_DS_−T) were obtained when V_DS_ was fixed at 2 V and V_G_ was fixed at −3 V.

## 3. Results and Discussion

In this work, the spacer image transfer technique (SIT) was chosen to fabricate the SiNW-FET biosensor image, as shown in Figure 1. In order to observe the final profile of the SiNW, the cross section of SiNW was characterized by transmission electron microscope (TEM), as shown in Figure 1a. As can be seen from the images, the width and height of the SiNW are 30.25 nm and 26.55 nm, respectively. Furthermore, the thickness of HfO_2_/SiO_2_ layers are 8.02 nm and 4.56 nm, respectively. The top layer is Au sprayed during the preparation of TEM samples. Figure 1b shows the EDS analysis of Si, Hf, O and Au elements. The results show that the HfO_2_ and SiO_2_ gate dielectric insulating layers are very homogeneous and the interface is clear and flat. The insulating layer is completely covered with silicon nanowires, which can reduce the leakage current from the liquid to the device and provide a stable liquid experimental environment. Appendix A shows the top view of the SiNW array sensor measured by SEM. As can be seen from the images, SiNW arrays were obtained using the SIT technique with high uniformity.

Figure 2a plots the drain current of SiNW-FET biosensor as a function of the biased gate voltage (I_DS_−V_G_). It can be seen from the figure that as the gate voltage (V_G_) changes from when 0 V changes to −20 V, the drain current increases sharply, and the device shows the characteristics of a typical PMOS field effect transistor. For the electrical characteristic curve of V_DS_ = 2 V, it is estimated that the switching ratio is greater than 10^6^ and the threshold voltage is about −7.38 V. Figure 2b shows the output characteristic curves (I_DS_−V_DS_) of the SiNW-FET sensor between the drain current (I_DS_) and the drain voltage (V_DS_, from 0 to 5 V) at different bias gate voltages V_G_ (from −20 V to 0 V). The drain current increases with the increase of V_G_ bias. The results show that the carrier concentration in SiNW can be adjusted linearly, and the device exhibits outstanding FET electrical performance.

The PEG can change the dielectric constant of the solution at interface, thereby increasing the effective Debye length (λ_D_), so as to achieve real-time detection of target molecules in high ionic strength solutions [20]. In addition, PEG can be used as a spacer to maintain the bioactivity of the recognition probe and can effectively adjust the density of the probe fixed on the surface of the biosensor, which is conducive to improving the sensitivity of the biosensor [21]. To detect glucose in high ionic strength buffers, we co-modified 3-(aminopropyl) trimethoxysilane (APTMS) and silane-PEG on the SiNW-FET surface, and then immobilized GOD on the SiNW-FET surface using GA as a linker molecule to construct the APTMS:PEG/GA/GOD/SiNW-FET glucose biosensor. The schematic diagram of the detection experimental device is shown in Figure 3. The electrical signal recorded by the biosensor in the biosensing measurement is obtained by the lock-in amplifier system [22]. The successful surface modification of SiNW-FET devices was demonstrated using AFM imaging (Appendix A), electrical characterization (Appendix A) methods and X-ray photoelectron spectroscopy characterization (Appendix A).

In the research of glucose sensing performance, the transfer characteristic curves (I_DS_−V_G_) of APTMS/PEG/SiNW-FET glucose biosensors modified with different ratios (APTMS:silane-PEG = 1:1, 1:2, 2:1) at different glucose concentrations were tested. Figure 4a–c shows that the I_DS_ of the three types of APTMS/PEG/SiNW-FETs decreased with increasing glucose solution concentration, resulting in a shift in threshold voltage (V_th_). This can be explained by the enzymatic reaction between glucose and GOD and the formation of products. The potential sensing mechanism of the glucose biosensor based on APTMS/PEG/SiNW-FET is attributed to the reaction of H_2_O_2_ generated by the oxidation of glucose. Glucose oxidase will catalyze glucose to produce gluconolactone and hydrogen peroxide (H_2_O_2_). Meanwhile, the hydrogen peroxide will dissociate to produce hydrogen ions (H^+^) and electrons (e^−^) [23]. When the gate voltage is −3 V, the SiNW-FET exhibits p-type semiconductor characteristics. Therefore, the number of carriers in the channel layer of SiNW-FETs decreases under H^+^ induction. It causes the current value to change with the change of glucose concentration [24]. The enzymatic reaction of glucose is as follows:(1)$$D–\mathrm{glucose}\;+{\;H}_{2}O\;+{\;O}_{2}\;\overset{\mathrm{GOD}}{\to }\;\mathrm{gluconolactone}\;+{\;H}_{2}O_{2}$$
H_2_O_2_ → O_2_ + 2H^+^ + 2e^− ^(2)

As shown in Figure 4d, the glucose content detected by the three APTMS/PEG/SiNW-FET glucose biosensors was plotted against the recorded threshold voltage offset (ΔV_th_) and obtained by linear fitting. The three calibration lines are compared, and the graph shows that all three types of APTMS/PEG/SiNW-FET glucose biosensors can recognize glucose in the concentration range of 100 nM~10 mM. Furthermore, a ratio of APTMS:silane-PEG = 2:1 could allow the APTMS/PEG/SiNW-FET to achieve the best limit of detection at 10 nM in 1× PBS. It can be determined that APTMS:silane-PEG = 2:1 is the optimal concentration ratio. This optimal ratio concluded on the APTMS/PEG/SiNW-FET surface provides (i) sufficient GOD density to ensure that it is compatible with the target binding of specific binding, (ii) increased effective Debye length by PEG for detection of glucose in high ionic strength solutions, and (iii) PEG can also be used as an interstitial molecule to maintain the bioactivity of the recognition probe and effectively regulate the density of the recognition probe fixed on the surface of the biosensor, which is beneficial to improve the sensitivity of the biosensor. 

In a further analysis, we used a simple planar packing model for the optimal ratio of APTMS/PEG ratio of 2:1. The model justified the 2:1 ratio as the optimal ratio. We assume that APTMS and GOD are covalently linked to each other. When the ratio of APTMS/PEG concentration is 2:1, we propose that the modification conforms to the following model, in which the diameter of GOD is 14 nm~18 nm [25], and the diameter of PEG is about 14 nm [26]. It can be seen from the Appendix A that under the condition of PEG interval modification, there are most GOD probes in this ratio. In this model, it obtains the most effective modification density when the ratio of APTMS/PEG is 2:1. The FET channel achieved the largest number of GOD surrounded by appropriate quantity of PEG due to the closed-packed arrangement. When the ratio of APTMS/PEG is 1:1, the modified GOD and PEG are equally distributed randomly. With this condition, GOD molecules can just fill in the gap of PEG when the ratio of APTMS/PEG is 1:2 (Appendix A). The PEG formation of stacks on the surface increases Debye screening length and effective modification density. Therefore, compared with the APTMS/PEG of 1:1, the glucose detection signal at 1:2 is better. This contribution tends to be weakened with the decrease of the number of GOD when the APTMS/PEG of 2:1 peaked as a result of the closed-packed arrangement model. Therefore, we prepared the APTMS/PEG/SiNW-FET devices with the ratio of APTMS:PEG = 2:1 for the subsequent experiments of detecting glucose in high ionic strength buffer solutions.

Figure 5a displays the drain current-time response using APTMS/PEG/SiNW-FET sensor at gate bias of −3 V. The measurement results show that when the glucose solution is added to the SiNW surface for less than 8 s, the generated current signal tends to be stable. This indicates that APTMS/PEG/SiNW-FET has a fast response to glucose detection. The absolute value of the source-drain current decreases as the glucose concentration gradually increases from 10 nM to 10 mM. The linear fitting curve between current value change and glucose concentration is shown in Figure 5b. The device shows an excellent linear relationship in the range of 10 nM~10 mM glucose concentration, the fitted linear relationship is y = 2.01 + 0.47x, and the linear correlation coefficient (R^2^) is 0.99747. The sensitivity of field-effect transistor biosensors is determined by the slope calculation of the linear range. In the range of glucose concentration from 10 nM to 10 mM, the sensitivity is 0.47 μAcm^−2^mM^−1^ and the detection limit is 10 nM. Table 1 shows the performance comparison between the “mixed catalytic layer “APTMS/PEG/SiNW-FET glucose biosensor and other previously reported FET-based glucose sensors. Compared with previous studies, our prepared APTMS/PEG/SiNW-FET is used as a new type of glucose sensor. The sensor has significant advantages for detecting a low concentration glucose solution. Compared with other types of GOD-based sensors that are not relying on FET, such as mesoporous films [27], electropolymerized films [28] and LbL films [29], the APTMS/PEG/SiNW-FET sensors have a fast response, wider detection range and lower detection limits.

After confirming the sensitivity of glucose, we investigated the specificity of the sensor by detecting the interference of uric acid (UA) and ascorbic acid (AA) on the “mixed-catalyzer layer” functionalized SiNW-FET chips. As shown in Figure 6a, there is no change in the current when other interfering substances are added in turn. However, the absolute value of current was reduced after adding the glucose solution. The signal response ΔI_DS_ of the APTMS/PEG/SiNW-FET biosensor is less than 0.25 μA when detecting the interference substances UA and AA, and about 2 μA when detecting glucose; the difference was at least eightfold (Figure 6b). The results indicated that the biosensor delivers excellent specificity between non-target and target glucose.

We have applied the APTMS/PEG/SiNW-FET sensor to detect glucose in serum samples. Different concentrations of serum glucose were selected for detection, and pure serum without glucose was used as a blank control test. As shown in Figure 7a, when the gate voltage is fixed, the source-drain current decreases with the increase of glucose concentration, and the detection limit is 10 nm. The signal response ΔV_th_ (Figure 7b) also increases gradually. Serum sample analysis shows that the APTMS/PEG/SiNW-FET biosensor has great application potential in detecting glucose in real samples.

## 4. Conclusions

In conclusion, an optimized method is proposed in our study to construct an APTMS/PEG/SiNW-FET glucose biosensor in a fast and efficient way by modifying the SiNW-FET surface with different ratios of APTMS and silane-PEG. The sensitive detection of low-molecular-weight glucose shows excellent response to glucose over a wide range of concentrations of 10 nM~10 mM when APTMS:silane-PEG is 2:1. The higher sensitivity is 0.47 μAcm^−2^mM^−1^, the low detection limit is 10 nM. Other advantages of our biosensor, such as short-time response (<8 s) and strong specificity, have been demonstrated in our research. In this work, the glucose biosensor using APTMS/PEG/SiNW FET has excellent application prospects in glucose detection, and the sensor is one of the candidates for the future manufacture of a real-time monitoring glucometer.

## Figures and Tables

**Figure 1 nanomaterials-13-00604-f001:**
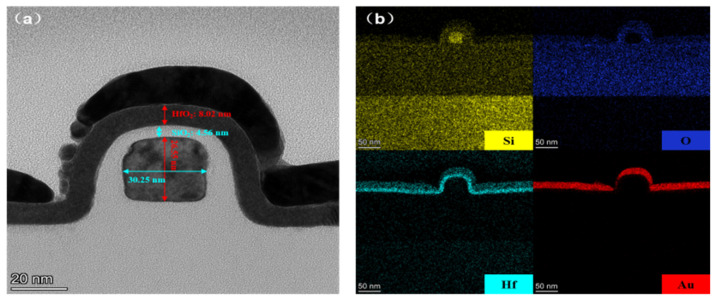
(**a**) TEM image of SiNW channel of the device. (**b**) EDS elemental mappings of Si, O, Hf and Au, respectively.

**Figure 2 nanomaterials-13-00604-f002:**
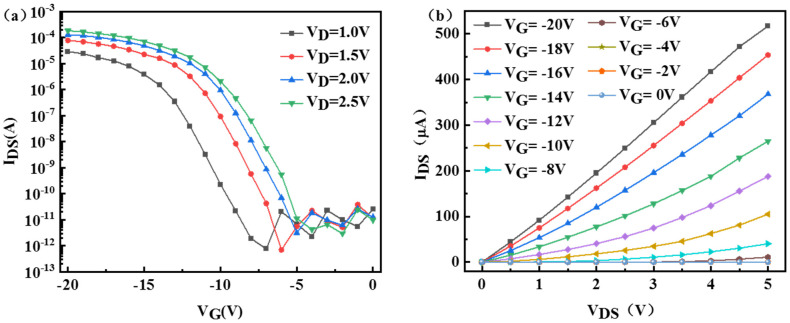
Electrical characteristic curve of SiNW−FET. (**a**) Transfer characteristic curve (I_DS_−V_G_). (**b**) Output characteristic curve (I_DS_−V_DS_).

**Figure 3 nanomaterials-13-00604-f003:**
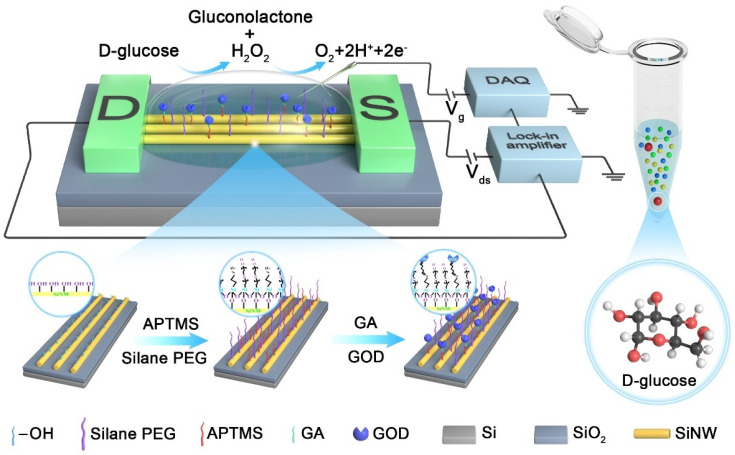
Schematic diagram of surface modification and glucose detection principle of APTMS/PEG/SiNW−FET glucose biosensor.

**Figure 4 nanomaterials-13-00604-f004:**
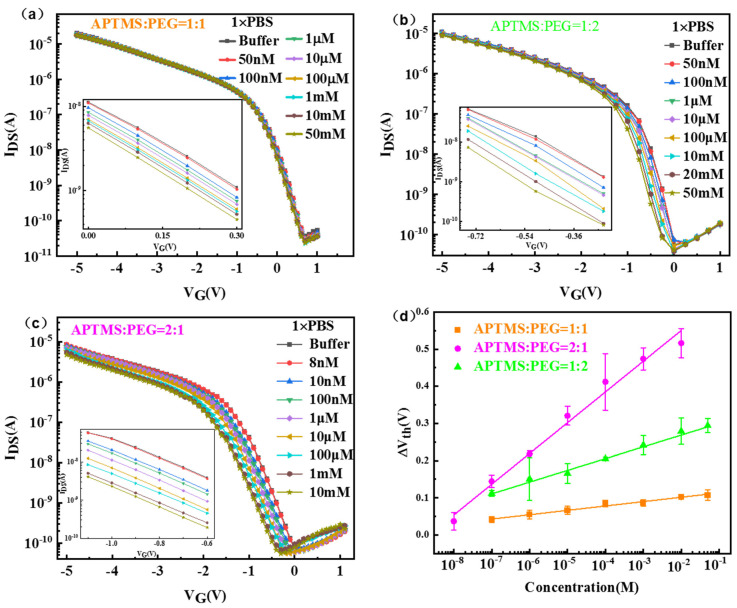
Electrical properties of three types of APTMS/PEG/SiNW−FET glucose biosensors for glucose detection. (**a**) APTMS:silane-PEG = 1:1. (**b**) APTMS:silane-PEG = 1:2. (**c**) APTMS:silane-PEG = 2:1. (**d**) The plot of the voltage offsets produced by the APTMS/PEG/SiNW−FET glucose biosensor and its corresponding calibration line as a function of glucose concentration. APTMS:silane-PEG = 1:1 (orange): The fitted linear relationship is y = 0.12 + 0.01x, and the linear correlation coefficient (R^2^) is 0.9855; APTMS:silane-PEG = 1:2 (green): The fitted linear relationship is y = 0.33 + 0.03x, R^2^ = 0.9903; APTMS:silane-PEG = 2:1 (purple): Fitted linear relationship y = 0.71 + 0.08x, R^2^ = 0.9978.

**Figure 5 nanomaterials-13-00604-f005:**
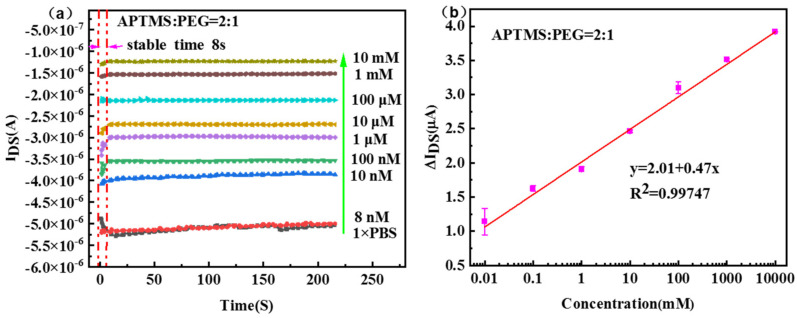
(**a**) Real−time response curve of glucose concentration in the range of 10 nM~10 mM for APTMS/PEG/SiNW−FET sensor when V_G_ = −3 V, V_DS_ = 2 V. (**b**) Calibration chart of steady−state current and glucose concentration, the relationship between current change and glucose concentration and its linear fitting curve, the fitted linear relationship is y = 2.01 + 0.47x, and the linear correlation coefficient (R^2^) is 0.99747.

**Figure 6 nanomaterials-13-00604-f006:**
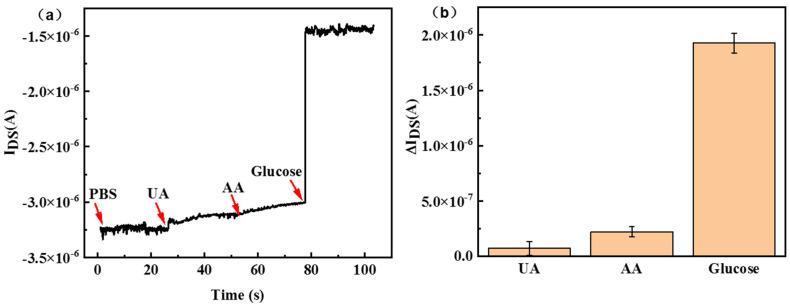
(**a**) Specificity test of APTMS/PEG/SiNW−FET glucose biosensor when 1 mM concentration of glucose and 50 mM concentration of UA and AA were added sequentially. (**b**) Corresponding current changes (ΔI_DS_).

**Figure 7 nanomaterials-13-00604-f007:**
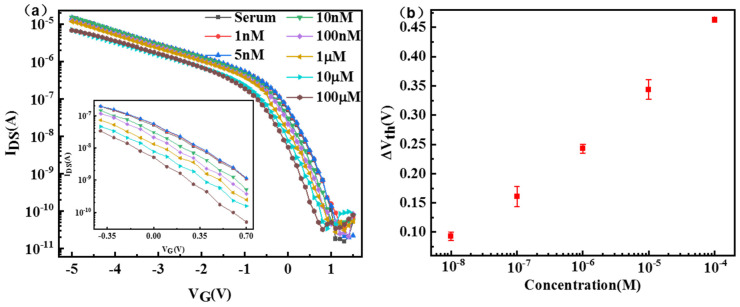
(**a**) Transfer curves and (**b**) the detailed ΔV_th_ of APTMS/PEG/SiNW−FET biosensors for detecting glucose in serum samples.

**Table 1 nanomaterials-13-00604-t001:** Performance of PEG and GOD co-modified SiNW-FET glucose biosensors compared to other FET-based glucose sensors.

Method	Detection Limit	Dynamic Range	Ref
GOx-ZnO NRs-FET	0.07 μM	0.07 μM~75 mM	Ahmad et al. [30]
Graphene-FET	1 μM	1 μM~10 mM	Kim et al. [31]
MoS_2_-FET	300 nM	300 nM~30 mM	Shan et al. [32]
IGZO-FET	170 μM	170 μM~30 mM	Du et al. [33]
GOD-GA-Ni/Cu-MOFs-FET	0.51 μM	1 μM~20 mM	Wang et al. [34]
PABA-graphene FET	4.1 μM	10 μM~1 mM	Fenoy et al. [35]
APTMS/PEG/SiNW-FET	10 nM	10 nM~10 mM	This work

## Data Availability

Not applicable.

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
