# Peer review of "Polyethylene Glycol Functionalized Silicon Nanowire Field-Effect Transistor Biosensor for Glucose Detection"

_nanomaterials, 2023, doi:10.3390/nano13030604_

Round 1

Reviewer 1 Report

The paper discusses the effects of using a PEG spacer to improve the detection of glucose by a silicon nanowire field effect transistor. The results shown compare favourable with those reported elsewhere in the literature. The authors report the results on a silane to peg ration of 2:1, 1:1 and 1:2. Perhaps surprisingly the worst performance is observed in the 1:1 device. While the 2:1 gives the best response the 1:2 is somewhere in between the two. You would expect a more sequential response in that the worst performance is that of the most PEG. However, I would like to invite the authors to comment on the observed trend in performance rather than just the statement that one of the ratios is the best performance, and provide a hypothesis on why this happens. Additionally, it raises the question, what would the performance be at other ratios?

There are a number of grammatical issues in lines 40, 56, 62, 82, 91, 96, 130 and 244. In general the fonts inside the figures is too small and Figure 3 is too small to be understandable.

Author Response

Reviewer #1: 1. The paper discusses the effects of using a PEG spacer to improve the detection of glucose by a silicon nanowire field effect transistor. The results shown compare favourable with those reported elsewhere in the literature. The authors report the results on a silane to peg ration of 2:1, 1:1 and 1:2. Perhaps surprisingly the worst performance is observed in the 1:1 device. While the 2:1 gives the best response the 1:2 is somewhere in between the two. You would expect a more sequential response in that the worst performance is that of the most PEG. However, I would like to invite the authors to comment on the observed trend in performance rather than just the statement that one of the ratios is the best performance, and provide a hypothesis on why this happens. Additionally, it raises the question, what would the performance be at other ratios?

Reply: Thans for these constructive opinions. We have a detailed description of this issue in Section 5 of the supporting information.

  1. There are a number of grammatical issues in lines 40, 56, 62, 82, 91, 96, 130 and 244. In general the fonts inside the figures is too small and Figure 3 is too small to be understandable.

Reply: Thanks for this good advice. We have modified the corresponding sentence grammatical issues in the text. Also enlarge the picture and the corresponding font, easy for readers to understand.

Reviewer 2 Report

The paper can be published after minor revision reflecting comments inserted as yellow notes into attached pdf of submitted manuscript and suplementary material.

Author Response

Reviewer #2:

The paper can be published after minor revision reflecting comments inserted as yellow notes into attached pdf of submitted manuscript and suplementary material.

Reply: Thanks for this good advice. We have made revisions per yellow annotations in the text.

Reviewer 3 Report

This manuscript from Wei, Luo et al introduces glucose sensors based on GOD molecules immobilized on PEG-grafted Silicon nanowire Field-Effect Transistors.  While the nanostructural control of the device is well demonstrated, there are issues that need to be adressed regarding the surface functionalization and the sensor's behavior. 

- Comments on the form of the manuscript:

1- The english language and style is good in the introduction part, but becomes in the Materials/Method and Results/discussion parts.

2-Figure 4 is not readable. PLease consider increasing the text size.

-  Comments on the scientific content:

3- The surface functionalization of the sensors is poorly characterized in terms of chemical composition. Performing XPS analysis (or QCM on similar surfaces) as well as ATR-FTIR are advised :

             - what is the density (or the quantity) of PEG chains grafted on the device. This information is critical as it will control the conformation of PEG chains and thus the access to the substrate.

              - How much GOD was immobilized on the sensor ?

                - Demonstration of the covalent immobilization of GOD should be provided. 

4- QUestions arising from the sensor's behavior:

             - Can the authors comment on the reusability and aging of the sensors after multiple detection cycles ?

               - As mentionned in the introduction, measurements in serum are challenging. Why this condition was not tested for the sensor ? Even moderate results under these conditions would greatly improve the impact of this work.

              - Why is the APTMS/PEG ratio 2:1 yielding the best results ? Could the authors discuss this point ?

5 - The bibliography can be improved on the following points:

       - Comparisson with other types of GOD-based sensors that are not relying on FET, such as mesoporous films  (https://doi.org/10.1007/s44211-022-00209-0), electropolymerized films (https://doi.org/10.1016/j.bios.2020.112408), LbL films (DOI10.1016/j.jelechem.2013.03.001)

        - The introduction shall mention other approaches to functionalize the surface of nanostructured devices including electrosynthesis (https://doi.org/10.1002/smll.201500639) and electrodeposition (https://doi.org/10.1246/cl.2012.383)

Author Response

Reviewer #3:

  1. The english language and style is good in the introduction part, but becomes in the Materials/Method and Results/discussion parts.

Reply: We have revised the English language and style of the Materials/methods and Results/Discussion sections of the text.

  1. Figure 4 is not readable. Please consider increasing the text size.

 Reply: We have enlarged the text size in Figure 4.

  1. The surface functionalization of the sensors is poorly characterized in terms of chemical composition. Performing XPS analysis (or QCM on similar surfaces) as well as ATR-FTIR are advised :

    - what is the density (or the quantity) of PEG chains grafted on the device. This information is critical as it will control the conformation of PEG chains and thus the access to the substrate.

    - How much GOD was immobilized on the sensor ?

    - Demonstration of the covalent immobilization of GOD should be provided.

Reply: (1). For the density problem of density of PEG, we propose a model in Section 5 of the support information.

          (2). For the number of modified GOD, we present a model in Section 5 of the supporting information and explain it in detail.

         (3). The proof of covalent immobilization of GOD is detailed in Section 4 of the Supporting Information.

  1. Questions arising from the sensor's behavior:

    - Can the authors comment on the reusability and aging of the sensors after multiple detection cycles ?

    - As mentionned in the introduction, measurements in serum are challenging. Why this condition was not tested for the sensor ? Even moderate results under these conditions would greatly improve the impact of this work.

    - Why is the APTMS/PEG ratio 2:1 yielding the best results ? Could the authors discuss this point ?

Reply: (1).Thanks for reviewer’s suggestion. In this paper, our purpose is to demonstrate the use of different ratios of APTMS/PEG to regulate the density of GOD for glucose detection. The performance of reusability and aging is the research content of our next stage of work.

        (2). Serum glucose testing is described in detail in the Results and Discussion section.

       (3). The reason APTMS/PEG ratio of 2:1 yields the best results is detailed in Section 5 of the supporting information.

  1. The bibliography can be improved on the following points:

    - Comparisson with other types of GOD-based sensors that are not relying on FET, such as mesoporous films  (https://doi.org/10.1007/s44211-022-00209-0), electropolymerized films (https://doi.org/10.1016/j.bios.2020.112408), LbL films (DOI10.1016/j.jelechem.2013.03.001)

    - The introduction shall mention other approaches to functionalize the surface of nanostructured devices including electrosynthesis (https://doi.org/10.1002/smll.201500639) and electrodeposition (https://doi.org/10.1246/cl.2012.383)

Reply: (1). The above references are cited in the Results and Discussion section [25], [26], [27] and discussed in the text

          (2). The above literature is cited in the introduction part [6], [7].
